# LLMs Synergy:
# From Closed-Source Prototyping to Open-Source Model based Instruction Following

## Abstract

We study the problem of constructing an efficient LLM-based instruction-following agent capable of comprehending and executing open-ended instructions in an embodied environment. We propose a method called LLMs Synergy for rapid domain adaptation in the instruction-following task without requiring additional manual annotations. This approach leverages a large general-purpose LLM to establish task baselines and generate domain-specific data. The knowledge from the larger model is then gradually transferred to a domain-tuned open-source LLM through a model transition process, enabling faster and more efficient adaptation. Accordingly, we developed the Dynamic Instruction Decomposition (DID) framework, specifically designed for LLM integration within this task scenario. The DID framework enables the agent to progressively align open-ended natural language commands with dynamic environmental contexts. Experimental results demonstrate significant improvements in task accuracy, leading to more effective instruction following and enhanced human-agent collaboration.

## 1 Introduction

Recent advancements in Large Language Models (LLMs) have marked a new phase in the development of AI agents (Yao et al., 2022; Wang et al., 2023; Huang et al., 2023; Dong et al., 2022), facilitating more natural human-agent collaboration. Traditional agents have often struggled to comprehend open-ended instructions within embodied environments. The enhanced natural language understanding and reasoning capabilities exhibited by LLMs offer promising solutions to these challenges.

In this paper, we address the challenge of incorporating LLMs to develop instruction-following agents within a collaborative research environment, specifically the CB2 scenario (Sharf et al., 2023). CB2 is a real-time collaborative framework in which a human leader and an agent follower cooperate to collect matching card sets in a shared 3D space. The primary difficulty arises from the differing perspectives of the human leader and the agent follower, compounded by the open-ended nature of the instructions. This often results in ambiguity, requiring the agent to autonomously interpret high-level guidance and translate it into concrete actions based on its observations.

While proprietary closed-source LLMs like Gemini 1.5 (Team, 2024), GPT3.5 Ouyang et al. (2022) and GPT4 (OpenAI et al., 2024), have demonstrated remarkable capabilities as general-purpose agents in solving a variety of tasks, their model opacity and limited accessibility hinder their applications in specific domains. In contrast, open-source, low-parameter LLMs, which offer reduced computational costs and greater adaptability, are often preferred for domain-specific tasks. However, these models typically face an initial performance gap when compared to proprietary models. The key challenge lies in efficiently enhancing the domain-specific performance of smaller LLMs. This can be achieved through (1) the design of

domain-specific execution frameworks that leverage LLM strengths, and (2) the acquisition of high-quality domain-specific data for fine-tuning. Notably, large, general-purpose closed-source models, owing to their superior generalization, can contribute significantly to both strategies.

We introduce **LLMs Synergy**, a novel approach to rapidly adapt large language models (LLMs) for domain-specific tasks, leveraging the complementary strengths of both proprietary closed-source and open-source models. Our contributions are as follows:

- We propose **LLMs Synergy** as a framework for efficient domain adaptation in instruction-following tasks. This method utilizes larger LLM to establish task baselines and generate domain-specific data. Subsequently, knowledge is progressively transferred through a model transition process to a domain-tuned open-source LLM, enabling faster and more efficient adaptation.

- We develop the **Dynamic Instruction Decomposition (DID)** framework, designed for embodied instruction comprehension and execution. DID incrementally aligns open-ended natural language instructions with dynamic environmental contexts, enhancing the ability of LLMs to understand and execute complex tasks through a progressive exploration-based decomposition of instructions.

- By integrating LLMs with the DID framework, we significantly improve agents' comprehension of natural language and their adaptability to dynamic environments. Furthermore, by leveraging smaller, task-specific open-source models, we reduce computational overhead while maintaining task accuracy. Our experimental results demonstrate substantial improvements in task performance, particularly in instruction-following and human-agent collaboration.

## 2 RELATED WORK

### 2.1 INSTRUCTION FOLLOWING

Recent advancements in instruction following for robotics have demonstrated notable progress, particularly in the comprehension and execution of specific commands (Huang et al., 2022; Ahn et al., 2022). Nonetheless, substantial challenges remain in handling open-ended instructions and dynamic environments, as these scenarios demand a deeper integration of real-world commonsense reasoning and the ability to process complex natural language instructions within context (MAV, 2015). Furthermore, existing instruction-following models are typically trained on narrowly defined tasks, limiting their generalization capabilities (Chen et al., 2023), and frequently lacking the contextual awareness required for robust decision-making (Wang et al., 2021). This issue is exacerbated by the reliance on large-scale annotated datasets (Shridhar et al., 2020; Misra et al., 2017; Suhr et al., 2018), which are both costly and labor-intensive to curate. In contrast, our approach mitigates these limitations by leveraging the broader generalization and generative capacities of large proprietary models, thereby reducing the dependency on extensive annotated data.

### 2.2 LLM-BASED AGENT

Many prior works have explored various methods for using frozen LLMs to build agents across different domains. A significant amount of research focuses on prompting techniques (Wei et al., 2022; Yao et al., 2024; 2022; Dong et al., 2022) to enhance the performance of large foundation models in specific domains. Among them, in-context learning, which incorporates feedback from the environment, is commonly used. For instance, Voyager (Wang et al., 2023) and Ghost(Zhu et al., 2023) iteratively prompt LLMs to regenerate action code based on error messages, continually refining the prompt with this information. Our approach differs from these prompting-based methods by conducting domain-specific fine-tuning at a lower cost, which leads to more robust and controllable model performance.

Additionally, some works have concentrated on supervised fine-tuning, for example, E2WM (Xiang et al., 2024) and LLAMARider (Feng et al., 2023) focus on collecting high-quality data to fine-tune LLMs. Specifically, E2WM gathers embodied experience in VirtualHome using Monte Carlo Tree Search and random exploration, while LLAMARider collects experience in the game engine Minecraft via self-reflection with feedback. Both approaches illustrate that fine-tuning on collected experiences significantly improves LLMs' capability to address tasks within their respective environments. However, these methods often necessitate substantial effort to amass environment-specific data due to initial model performance limitations, thereby requiring extensive searches for high-quality data. Our approach mitigates this challenge by employing larger models to swiftly establish a baseline, facilitating the efficient accumulation of data to guide the fine-tuning of smaller models.

## 3 COLLABORATIVE ENVIRONMENT

**Overview** In this study, we explore human-agent collaboration in the CB2 scenario Sharf et al. (2023), a real-time collaborative environment where a human leader and an agent follower work together to collect matching card sets in a shared 3D space.

A valid set comprises three cards, each differing in color, shape, and count. When the selected cards in the environment form a valid set, the players are awarded a point. The game is turn-based, with a limited number of total turns. The leader plans and provides instructions in natural language, while the follower executes these commands. The goal is to maximize the final score by successfully collecting card sets through effective collaboration, where the task score reflects the efficiency of the cooperation. More details of the environment are provided in Appendix A.

This game involves two key aspects: the Observability Gap and the Ability Gap. The leader has an overhead view of the environment, while the follower sees only from a first-person perspective, with some card patterns hidden. As a result, the descriptions within leader's instructions focus on the surroundings rather than specific details, such as "the card near the stone" or "the card between the lake and the yellow house". Additionally, the follower has greater mobility, covering more ground per turn, making task success mainly dependent on the follower's ability to correctly execute each instruction.

CB2's design effectively captures real-world collaborative challenges. Differences in perspective and the use of open-ended language can lead to ambiguous or unclear instructions, sometimes with inaccuracies. This necessitates that the follower autonomously interpret and translate high-level instruction into concrete actions based on their observations.

**Data** The CB2 research team also released a dataset of human-human interactions, where trained human workers excel as leaders and followers. This dataset includes leader instructions, follower actions, final card selections, and game states including map information and final scores. The data is divided into two parts: training and evaluation sets. The training data, consisting of 185 games and 3,439 instructions, can be used for model development, while the evaluation set of 187 games and 3,404 instructions is used for comparing agent's performance. Each instruction is recorded with the human leader's instruction $x$, follower's first-view map $map$, the follower's position $pos$ and the states of all cards $C$ during execution.

## 4 METHOD

This section outlines the synergy between a general LLM and a task-specific LLM for effective instruction following. To address the challenge posed by the different perspectives between the leader and follower, we first design an execution framework to embed LLM that enhances instruction comprehension and execution during dynamic environmental exploration. The general LLM is used to test the framework during the setup phase. Unpon framework completion, we will fine-tune a domain-specific smaller LLM to replace the

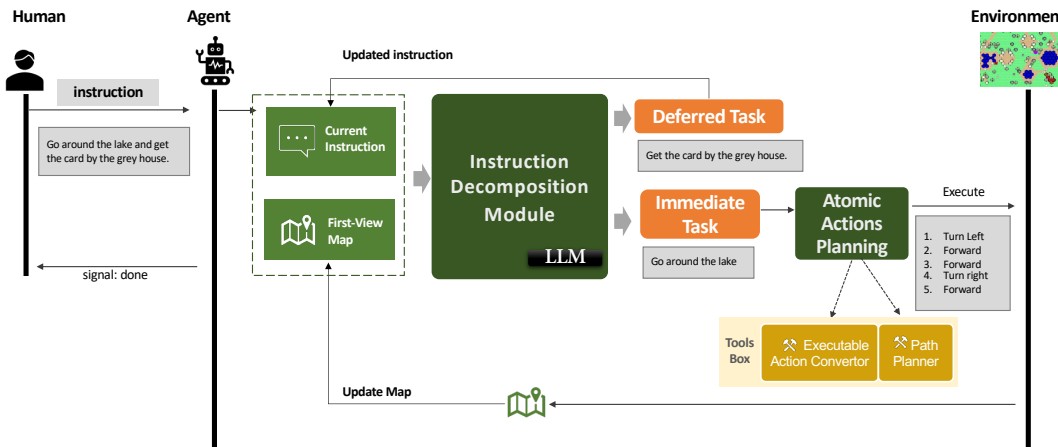

Figure 1: Synergizing LLMs Workflow. It illustrates the various stages of model transition and the respective roles of the two LLMs. The two icons on the far left represent a general-purpose large LLM and a task-specific smaller LLM. The labels "Follower" and "Leader" attched to the icons in the workflow indicate the roles the LLMs play during the collaborative task.

Figure 2: Dynamic Instruction Decomposition Framework. This figure outlines the key modules and the process from receiving a human leader's instruction to the agent signaling task completion.

general model without performance reduction.The whole transfer process is illustrated in the Figure 1. The following subsections outline the role of each model and the specific approach taken.

## 4.1 DYNAMIC INSTRUCTION DECOMPOSITION FRAMEWORK

To address the discussed challenges in embodied instruction grounding and execution, we design the Dynamic Instruction Decomposition (DID) framework. It leverages the LLM's language comprehension to decompose instructions and dynamically align them with the evolving environmental context through progressive exploration. A domain-specific prompt $P_F$ (detailed in next section) is designed to guide the LLM in breaking down the original instruction into two types of tasks:

- **Immediate Tasks**: Tasks can be completed right away within the current contexts
- **Deferred Tasks**: Tasks require a change in perspective to gain additional information.

Figure 2 illustrates the entire process of instruction decomposition. When the Leader issues a new instruction, the instruction, together with a structured text sequence representing the current first-person view, is fed to the LLM. The LLM then decomposes the instruction into Immediate Task and Deferred Tasks based on the current input. Next, the agent executes the immediate task in the environment and updates the first-person view map. If any deferred task remains, it will be input to the LLM along with the updated map for the next decomposition. The agent iteratively repeats the decomposition and execution process until all tasks are completed. The initial testing with LLMs demonstrated promising results in the decomposition of open-ended instructions. However, the decomposed tasks sometimes can not be converted into executable actions in the environment. While LLMs excel in reasoning, they may lack precision in tasks such as numerical calculations and format-specific mapping. Therefore, leveraging the LLM's function-calling capability, we equipped the agent with tools for accurate map-based action generation. These include: A Path Planner to generate movement sequences (e.g., 'Forward, Turn left') to guide the agent towards a specific position and an Executable Action Converter to transform other types of tasks into executable actions. The implementation details of these two tools can be found in Appendix C. The agent's execution process is also outlined in Algorithm 1. A vivid example of the dynamic instruction decomposition process is shown in Appendix D.

Through rapid adjustments and testing of the framework, even with a small subset of training samples, the LLM-enhanced framework has already demonstrated competitive performance compared to traditional methods that rely on extensive data. Unlike traditional behavioral cloning models, Our LLM-based instruction follower agent clearly demonstrates agent characteristics: it operates on open-ended instructions and goals, reasons about them, formulates plans, utilizes tools and interacts with dynamic environments.

---

**Algorithm 1:** Dynamic Instruction Decomposition Process

---

**Input:** Leader's instruction $x$, Follower's first-view map $Map$, Step limit $L_{step}$

**Data:** Current instruction $x'$, Current follower's first-view map $Map'$, Immediate Task $x_I$, Deferred Task $x_D$, Atomic actions Set $S_A$, Step counter $C_{step}$

Initialization: $x' \leftarrow x$ ; $Map' \leftarrow Map$ ; $C_{step} \leftarrow 0$ ;

**while** $x' \neq NULL$ and $C_{step} < L_{step}$ **do**

    **while** *True* **do**

        $x_I + x_D \leftarrow$ Decompose$(x', Map')$ ;

        **if** *pass the Self-Checking* **then**

            break;

        **end**

    **end**

    $S_A \leftarrow$ Get atomic actions of $x_I$ using Tools Box;

    Follower interact with the environment;

    **foreach** *action in $S_A$* **do**

        **if** $C_{step} < L_{step}$ **then**

            Execute action;

            $C_{step} \leftarrow C_{step} + 1$

        **else**

            break;

        **end**

    **end**

    $Map' \leftarrow$ Get map of the new perspective in the environment ;

    $x' \leftarrow x_D$ ;

**end**

---

## 4.2 MODEL TRANSITION

In this section, we will demonstrate how to iteratively fine-tune a domain-specific small-scale LLM by gradually transferring knowledge from a larger general-purpose model. The larger model would play a crucial part in the construction of the fine-tuning dataset, helping to eliminate the need for additional manual annotations.

**Basic Dataset Construction** The previously developed agent follower, a general-purpose LLM integrated into the DID framework, is used for data labeling by generating decomposition outputs $(y_I, y_D)$ given an instruction and the according first-view map $(x, map)$. The training set in CB2 would be served as input dataset $X$. Specifically, the game state for each instruction is loaded to initialize the environment, where the agent follower decomposes the instruction and interacts with the environment until the task is completed or the step limit is reached, and intermediate data tuples $(x, map, y_I, y_D)$ are collected. Human follower performance is then used as the ground truth to evaluate the agent's execution and filter out invalid data. By comparing card states post-execution, only matching intermediate data is retained, forming the basic training dataset $D_1^{lab}$ during the process. This process can be formulated as follows:

$$D_1^{lab} = Filter_1(\{x, map, y_I, y_D \mid (x, map) \sim X, (y_I, y_D) \sim p_g((y_I, y_D) \mid P_F \oplus (x, map))\})$$

**Dataset Expansion** However, the scale of data collected is quite limited owing to the number of original instructions and the general-purpose model' success match rate. To address this, we utilized the general-purpose model's generative capabilities to act as a Leader and generate more data to expand the dataset. Another specific prompt $P_L$ is designed to describe the Leader's task, instructing the model to imitate a human leader and generate a series of human-like instructions (details of this prompt can be found in Appendix B). Unlike the previous labeling process, where the input $(x, map)$ is sampled from the existing dataset, only the $map$ is sampled from the game engine, both $x$ and $(y_I, y_D)$ would be generated by the general-purpose LLM. At this stage, a simple quality controllerwhich is just a format checker is implemented since there were no human execution results for comparison, which only ensures key task elements like cards or positions matched the map. This verification ensured only a basic level of quality control, leaving room for improvement in addressing certain limitations of the process. This process can be formulated as follows:

$$D_1^{gen} = Filter_2(\{x, map, y_I, y_D \mid map \sim M, x \sim p_g(x) \mid P_L \oplus map, p_g((y_I, y_D) \mid P_F \oplus (x, map))\})$$

Through the above generation process, a more diverse dataset $D_1^{gen}$ was created. The datasets $D_1^b$ and $D_1^{gen}$ are merged into the dataset $D_1$, used to fine-tune the smaller, domain-focused model for the first iteration. This dataset equips the smaller model to handle domain-specific tasks effectively, and after training, its performance approaches that of the closed-source general-purpose model.

**Dataset Optimization** The dataset still includes some inefficient or unreasonable decomposition cases that cannot be filtered out by comparing execution results alone. With two performance-matched models, we can now synergize them to generate higher-quality instruction decomposition data. Similar to the basic dataset construction, we integrate the smaller domain-specific model to act as an agent follower for instruction decomposition. The larger general model then serves as a quality controller, semantically verifying the decomposition results. After this step, only the data where both models agree on the decomposition will be retained. The filtered data forms the dataset $\mathbb{D}_2^b$. To ensure diversity, we combine $\mathbb{D}_1^{gen}$ and $\mathbb{D}_2^b$ into $\mathbb{D}_2$, for the second round of model fine-tuning.

This process can be formulated as follows:

$$D_2^{lab} = Filter_{LLM}(\{x, map, y_I, y_D \mid (x, map) \sim X, (y_I, y_D) \sim p_d((y_I, y_D) \mid P_F \oplus (x, map))\})$$

Leveraging this enhanced dataset, the smaller model exceeds the baseline set by larger models, providing higher accuracy with reduced computational overhead, thus completing the model transition.

## 5 EXPERIMENTAL SETUP

**Training** For the general LLM, various top-tier proprietary models were tested during the domain task framework design (performances of different models embedded with our framework is shown in Table 6.2). Ultimately, the best-performing model, Gemini 1.5 Flash (Team, 2024), was selected as the general-purpose LLM for our method. Then we employ the mistral-v0.3-7b model (Jiang et al., 2023) as our domain-specific smaller LLM, and further use a 4-bit quantized version for improved speed. Each model is trained using LoRA (Hu et al., 2021) with $r = 32$, $\alpha = 32$. We use 8-bit Adam with a total batch size of 32 and a learning rate of 2e-4, and the seed is set to 3407. We train 5 epochs for each iteration of the model.

**Domain-Specific Prompt Design** The prompt template $P_F$ is designed to guide LLM as a follower to generate desired decomposition data. The instruction and first-view map pair $(x, map)$ would be further embedded in the input alongside the task description. Our desired output $(y_I, y_D)$, is followed by the keywords "Immediate Task" and "Deferred Task".

---

**Domain-Specific Tuning Data Template**

**Prompt as Follower** $P_F$

Below is a task description, paired with an input that provides further context. Write a response that appropriately completes the request.

Task description:

You would be provided with an instruction and a structured string that describes your first-view map. Your task is to break down the original instruction into two categories based on the map:

1. Immediate Tasks: Tasks that are achievable within your current perspective and can be completed immediately.

- Type 1: Change Direction

- Type 2: Move to a specific location in the first-view map

- Type 3: Interact with a Card at a Specific Location in the first-view map

2. Deferred Tasks: Tasks that necessitate a change in perspective or additional insights to be accomplished. If there are no deferred tasks, record the output as "NULL".

Provide your answer in JSON format with the following keys: Immediate Task, Deferred Task. Other formats are not accepted. Expected Output Format:

{

"Immediate Task": One of the three immediate tasks,

"Deferred Task": "NULL" or a consice description of the remaining instructions in no more then 20 words.

}

Here is the instruction and the according map:

  Instruction: $x$

  First-view Map: $map$

- - - - - - - - - - - - - - - - - - - - - - - - - - - - - - - - - - - - - - - - - -

**Formatted text output** $(y_I, y_D)$

{"Immediate Task": One of the three immediate tasks,

"Deferred Task": "NULL" or a concise description of the remaining instructions in no more than 20 words. }

---

# 6 RESULTS AND ANALYSIS

## 6.1 EVALUATION METRICS AND DATA

**Metrics** The agent's performance is evaluated against human performance at the instruction level using the evaluation dataset of human-human game records released along with the CB2 platform. Mean card state accuracy is assessed by comparing the final states of the cards between the human and agent, with only exact matches considered as correct. We also compare the mean distance of the final position between human followers and the agent across all instructions. This metric complements the card state accuracy by assessing the efficiency of movements.

**Data** However, directly using all human follower execution results from the evaluation set as ground truth for comparison may lead to certain errors, as some instructions may have poor execution outcomes even by human followers. To enhance evaluation data quality, the poorly executed instructions would be removed automatically based on two criteria: those canceled by the leader during execution and those with no changes in the card set before and after execution. This filtering process resulted in the CB2-Eval dataset, with 1,417 remaining test instructions from 109 games. However, during the evaluation, we observed that the filtered dataset still has has some limitations. There are two types of issue: the follower selected a card but did not do so correctly, and the leader's instructions were unclear or ambiguous to understand. The automated rules could not filter out these errors. To ensure rigor in the evaluation, we further filtered out these problematic instructions and constructed a higher-quality evaluation dataset with 786 instructions from 97 games, CB2-Eval-Filtered. This dataset will also be released to facilitate more objective and accurate evaluations.

To ensure objectivity in our evaluations, we will report results on both datasets, CB2-Eval and CB2-Eval-Filtered, when comparing with other instruction-following methods. For internal comparisons, as CB2-Eval-Filtered is more reliable, we will conduct a more detailed analysis only on this dataset.

## 6.2 COMPARISIONS TO OTHER INSTRUCTION FOLLOWING METHODS

We compare our method with the behavior cloning model DT Sharf et al. (2023), which uses Decision Transformer as its architecture, as well as the GTPfollower[1] embedded in the CB2 Platform, which is developed with GPT3.5 Turbo. We also used the designed DID framework to integrate various LLMs for building more LLM-based agent followers. In addition to the general-purpose LLM Gemini1.5 Flash (Team, 2024) and the Mistral 7b (Jiang et al., 2023) used in our method, we also apply the DID framework to GPT3.5 Turbo (Ouyang et al., 2022) as GPTFollower, for comparison with our final approach. Table 6.2 provides a detailed comparison of all the methods.

Our method achieves the highest instruction execution accuracy on both datasets, with only a slight margin behind DID-Gemini in the average distance metric in CB2-Eval, yet significantly outperforming other approaches. Additionally, the agents combining top proprietary models with the DID framework show promising performance compared to DT and GPTfollower. Particularly, when compared to GTPFollower, which also uses the same GPT3.5 Turbo base, the superiority of the DID framework is clearly demonstrated. For open-source, low-parameter LLM pretrained Mistral-7b, directly integrating its pretrained version into the DID framework initially showed a significant performance gap compared to proprietary counterparts. However, after fine-tuning and incorporating the DID framework, the 7b model outperformed commercial models such as GPT3.5 Turbo and Gemini1.5 Flash, showcasing the effectiveness of our domain-specific fine-tuning approach.

---

[1]https://github.com/lil-lab/cb2/tree/main/src/cb2game/agents

| Level | CB2-Eval | | CB2-Eval-Filtered | |
|---|---|---|---|---|
| | Acc. ↑ | AvgDis ↓ | Acc. ↑ | AvgDis ↓ |
| DT | 30.37% | 3.18 | 40.09% | 2.22 |
| GPTFollower | 15.76% | 3.32 | 19.63 % | 2.67 |
| DID-GPT | 18.35% | 2.93 | 32.34 % | 2.22 |
| DID-Gemini | 39.31% | **2.30** | 53.29% | 1.63 |
| DID-Mixtral-7b-Pretrained | 2.61% | 3.58 | 3.09 % | 2.93 |
| DID-Mixtral-7b-Finetuend(Ours) | **40.71%** | 2.42 | **55.91%** | **1.57** |

Table 1: Comparisons of different agent followers on instruction execution accuracy and average distance across the two evaluation datasets. The GPT-based method utilizes GPT3.5 Turbo, while the Gemini-based approach is implemented using Gemini 1.5 Flash.

A further comparison of the model's performance on both datasets shows that metrics improved on the cleaned dataset, confirming that most of the removed data was indeed problematic. The relative ranking of the models remains almost identical, further validating the accuracy of our data cleaning process.

### 6.3 ABLATION STUDY

**Effectiveness of DID** It is worth noting that the small-scale LLM Mistral-7b, without domain-specific training, performed poorly due to its inherent limitations, regardless of the inclusion of the DID framework. Therefore, it is not suitable for assessing the framework's effectiveness. Instead, we use the general-purpose models for validation. We selected two leading closed-source LLMs, GPT3.5 Turbo and Gemini1.5 Flash, as the base models. We replaced the LLM in GPTFollower with Gemini, creating GeminiFollower, and compared it with DID-Gemini. The results in 6.3 show that, using the same base model, the DID framework brings significant performance improvements. The comparison between the two versions of GPT3.5 Turbo further validates the effectiveness of the DID framework. Notably, as Gemini1.5 Flash was used during the framework's design and testing phase—with prompts and tools tailored to its specific characteristics, the DID framework provides more significant gains for Gemini1.5 Flash compared to GPT3.5 Turbo.

**Impact of Dataset Variations** The fine-tuning with domain-specific data generated under the DID framework brought a significant improvement to Mistral 7b. We then further analyzing the model's performance of the model transition phase based on the training over various dataset versions. This includes the basic dataset, data generated by the general-purpose model, and data created through the synergy of both models. As shown in the table, the model's performance improved progressively as different data types were introduced.

In the first stage, the foundational dataset brought the largest performance gain as the smaller LLM adapted quickly to the DID framework's format. However, there remained a noticeable gap between the model and the performance of the general-purpose LLM. In the next stage, the more diverse data generated by the general-purpose model helped narrow this gap. Notably, in the final stage, with the addition of high-quality data combining the strengths of both models, performance increased again, and the smaller model ultimately surpassed the general-purpose model.

## 7 DISCUSSIONS

Beyond its research significance, the potential applications of human-machine collaboration are vast, warranting deeper exploration. However, our agent currently operates by passively receiving and executing instructions, which limits its capabilities and hinders its ability to take on more tasks. For broader future

| | CB2-Eval-Filtered | | | |
|---|---|---|---|---|
| | Acc.↑ | Diff. | AvgDist.↓ | Diff. |
| w/o DID (GeminiFollower) | 18.47% | | 2.79 | |
| DID-Gemini | 53.29% | 34.82%↑ | 1.63 | 1.16↓ |
| w/o DID (GPTFollower) | 19.63% | | 2.67 | |
| DID-GPT | 32.34% | 12.71%↓ | 2.22 | 0.45↓ |
| w/o Basic Data | 3.09% | | 2.93 | |
| w/o Generated Data | 46.24% | 40.86%↑ | 1.92 | 1.01↓ |
| w/o Dataset Optimization | 53.10% | 6.86%↑ | 1.66 | 0.26↓ |
| Ours | 55.91% | 2.81%↑ | 1.57 | 0.09↓ |

Table 2: Evaluation statistics on accuracy of instruction execution and average distance on the CB2-Eval-Filtered dataset. To evaluate the DID framework, performance differentials are computed across frameworks using the same base LLM. When assessing the impact of dataset variations, differentials reflect performance before and after incorporating the new data.

collaboration scenarios, it is crucial to enhance the agent's proactivity and its interaction with the leader, enabling it to participate in planning and thus improve collaboration efficiency.

Future research can advance this field by focusing on the following areas: First, developing more expressive feedback mechanisms, such as natural language and bidirectional dialogue, can greatly enhance system performance despite added complexity. Additionally, enhancing the autonomy of agents as followers by integrating them into holistic strategic planning with the leader. This would allow agents to provide valuable insights and recommendations, therefore improve collaboration success rates.

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

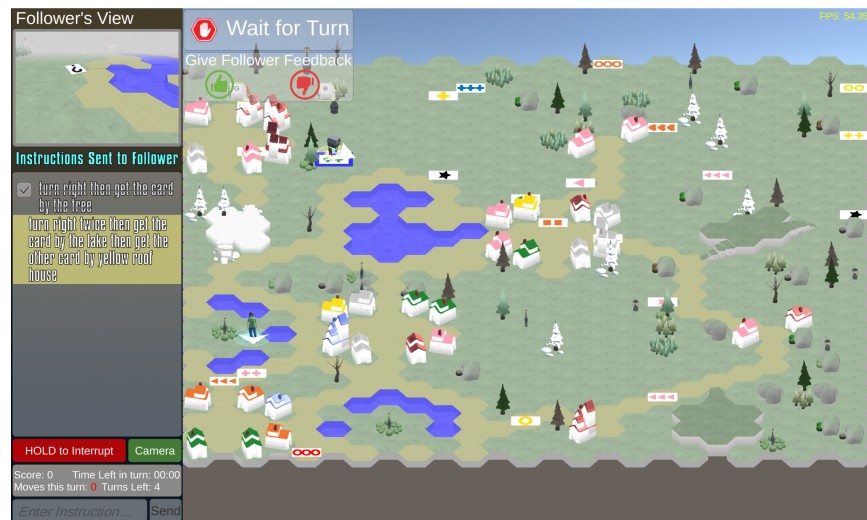

Figure 3: The interface of a human leader. The Leader's view is the complete environment that takes the main part of the image, while on the top left is a Follower's view.

Shunyu Yao, Dian Yu, Jeffrey Zhao, Izhak Shafran, Tom Griffiths, Yuan Cao, and Karthik Narasimhan. Tree of thoughts: Deliberate problem solving with large language models. *Advances in Neural Information Processing Systems*, 36, 2024.

Xizhou Zhu, Yuntao Chen, Hao Tian, Chenxin Tao, Weijie Su, Chenyu Yang, Gao Huang, Bin Li, Lewei Lu, Xiaogang Wang, et al. Ghost in the minecraft: Generally capable agents for open-world environments via large language models with text-based knowledge and memory. *arXiv preprint arXiv:2305.17144*, 2023.

## A    DETAILS OF CB2

The collaboration in CB2 interaction involves two agents: a leader and a follower, who work together to complete tasks but differ in their environment observations and abilities. Both agents can move between adjacent hexagons or turn in place to change orientation. They interact with cards by moving over them to select or deselect .The goal is for the agents to select valid sets of cards. A valid set consists of three cards with unique combinations of color, shape, and count. When a valid set is selected, the cards disappear, and the agents earn one point. Three new random cards appear in random positions, and the agents receive extra turns, though the number of additional turns decreases after each set completion.

As shown in Figure 3 The leader has a full overhead view of the environment, while the follower only sees what's directly ahead from a first-person perspective. Initially, the patterns on unselected cards are hidden from the follower, showing a question mark instead. The agents take turns, with each turn allowing a limited number of steps. Every movement (forward, left, right, or backward) consumes one step. Turns are time-limited to keep the interaction dynamic and minimize waiting for the other agent. The time limit can be adjusted, but typically, the leader is given more time to plan their moves. Turns alternate between the follower and the leader. Instruction writing and sending by the leader, and marking them as complete by the follower do not consume steps

# B PROMPT TO ACT AS A LEADER

The prompt template $P_L$ that use to guide LLM as a Leader to generate desired human-like instruction is listed below. The first-view map ($map$) would be embedded in the prompt.

---

**Domain-Specific Tuning Data Template**

**Prompt to Act as a Leader** $P_L$

Below is a task description, paired with an input that provides further context. Write a response that appropriately completes the request.

Task description:

You are a commander in a strategy game, responsible for providing clear and concise movement instructions to a follower. Your instructions should be structured into two parts: Immediate Task and Deferred Task, which means the instruction can be broken down into two steps. Your instructions should guide the follower to explore the map effectively and efficiently. Generate a variety of movement commands that direct the follower. Use human-like language and diverse phrasing, utilizing the landmarks and terrain features mentioned in the map. Additionally, must not include the corresponding location or interacting card's details derived from the Map Information in each command. Don't use the "tile" description in your instruction cause it's not human-like language. You should provide a list of instructions that fully utilizes the Map Information. Make sure that the instructions are varied and natural-sounding and the types of instructions are evenly distributed.

1. Immediate Tasks: Tasks that are achievable within your current perspective and can be completed immediately.

- Type 1: Change Direction
- Type 2: Move to a specific location in the first-view map
- Type 3: Interact with a Card at a Specific Location in the first-view map

2. Deferred Tasks: Tasks that necessitate a change in perspective or additional insights to be accomplished. If there are no deferred tasks, record the output as NULL.

Here is the first-view map: $map$

Provide your answer in JSON format with the following keys: Instruction, Immediate Task, Deferred Task. Other formats are not accepted. Expected Output Format:

{ "Instruction": instrucition text,

"Immediate Task": a task within the three types,

"Deferred Task": "NULL" or a consice description of the remaining instructions in no more then 20 words.

}

- - - - - - - - - - - - - - - - - - - - - - - - - - - - - - - - - - - - - - - - - - - - - - - - - - -

**Formatted text output** $(y_I, y_D)$ {

"Immediate Task": One of the three types,

"Deferred Task": "NULL" or a consice description of the remaining instructions in no more then 20 words.}

---

# C TOOLS IN DID

This section presents the implementation logic of the two tools, Path Planner and Executable Action Converter, within the DID framework.

---

**Algorithm 2:** Path Planner Algorithm

---

**Input:** $immediate\_task$, $follower\_location$, $map$, $cards\_location$
**Output:** $atomic\_actions$

**Function** get_target_location($immediate\_task$)**:**
    pattern $\leftarrow$ 'Tile at heading [-] and distance [-]:  [-]';
    target_locations $\leftarrow$ findall(pattern, $immediate\_task$);
    **return** $target\_locations$;

$current\_location \leftarrow follower\_location$;
$target\_locations \leftarrow$ get_target_location($immediate\_task$);
$action\_string \leftarrow$ "";
**foreach** $target\_location$ **in** $target\_locations$ **do**
    $action\_string \leftarrow action\_string+$ deep_first_search($current\_location$, $target\_location$,
    $map$, $cards\_location$);
    $current\_location \leftarrow target\_location$;
**end**
**return** $action\_string$

---

**Algorithm 3:** Excutable Action Convertor Algorithm

---

**Input:** $response\_dict$, $map$, $prop\_update$, $follower\_location$
**Output:** $action\_string$

$deferred\_task \leftarrow response\_dict["DeferredTask"]$;
$immediate\_task \leftarrow response\_dict["ImmediateTask"]$;
$action\_string \leftarrow$ "";
**if** *"Change Direction" or "Move"* **in** $immediate\_task$ **then**
    $action\_string \leftarrow$ immediate_task.split(":")[1].strip();
**end**
**else**
    $action\_string \leftarrow$ path_planner($immediate\_task$, $follower\_location$, $map$, $cards\_location$);
**end**
**if** $deferred\_task == "NULL"$ **then**
    $action\_string \leftarrow action\_string + ", done"$;
**end**
**return** $action\_string$

---

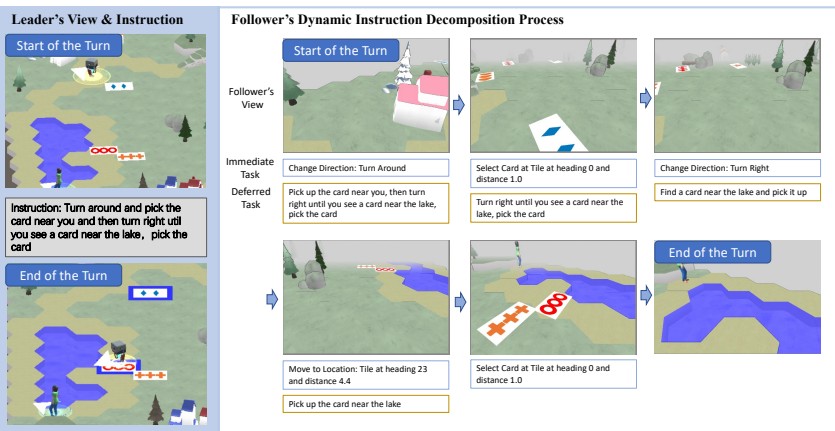

Figure 4: The interface of a human leader. The Leader's view is the complete environment that takes the main part of the image, while on the top left is a Follower's view.

## D  DECOMPOSITION EXAMPLE UNDER DID

Figure 4 illustrates an example of the dynamic instruction decomposition process. Upon receiving an instruction from the leader, the follower autonomously decomposes the instruction based on its current environmental context. Through a dynamic execution process, the follower continuously acquires new perspectives, enabling further instruction execution and ultimately achieving the overall goal set by the leader.

