# OpenReview forum: "LLMs Synergy : From Closed-Source Prototyping to  Open-Source Model  based  Instruction Following"
_ICLR.cc/2025/Conference — ICLR 2025 Conference Withdrawn Submission_

### Official Review · Reviewer_c16S · 2024-11-04

**Soundness:** 2
**Presentation:** 2
**Contribution:** 2
**Rating:** 3
**Confidence:** 3

**Summary:**

This paper presents LLMs Synergy, an approach which harnesses the capabilities of large language models (often proprietary) to generate domain-specific data for training specialized, open-sourced LLMs; and Dynamic Instruction Decomposition (DID), a framework that aligns natural language instructions with a target environment and breaks them down into executable steps. The authors evaluate the performance of these frameworks in a CB2 environment, reporting preliminary results that demonstrate the effectiveness of such an approach.

**Strengths:**

- The proposed LLMs Synergy approach enables the effective utilization of large proprietary LLMs to augment smaller, fine-tuned LLMs by generating additional high-quality training data and implementing quality control measures.
- The DID framework proposed in this paper demonstrates a strong affinity with the CB2 environment, and its efficacy is convincingly reported through the experiment results.
- The experimental evaluation is underpinned by a meticulously curated dataset, which undergoes rigorous evaluation, filtering, and refinement to ensure exceptional quality. This careful attention to detail contributes significantly to the reliability and validity of the reported results.

**Weaknesses:**

- The LLMs Synergy approach lacks novelty to meet the bar of ICLR -- the idea of ‘using large LLM to generate data to train smaller LLM’ approach has already been studied and researched within the community intensively over the past few years, see this survey [1] for some related papers.
- The experiments are conducted only on the CB2 environment, which may not be sufficiently representative or persuasive. To enhance the validity and generalizability of the findings, additional results from similar environments or tasks should be reported, e.g. Habitat 3.0 [2]

[1] Xu, X., Li, M., Tao, C., Shen, T., Cheng, R., Li, J., Xu, C., Tao, D. and Zhou, T., 2024. A survey on knowledge distillation of large language models. arXiv preprint arXiv:2402.13116.

[2] Puig X, Undersander E, Szot A, et al. Habitat 3.0: a co-habitat for humans, avatars and robots. arXiv preprintarXiv:2310.13724, 2023.

**Questions:**

The writing of this paper could be improved. The current method section is quite lengthy and it is quite hard to grasp the essential details. A clearer and more succinct explanation of the methodology would enhance the overall readability and comprehension of the paper.

Also there are several typos in the paper, please fix:
- 'Unpon framework completion, we will fine-tune a...'
- 'At this stage, a simple quality controllerwhich is just a format checker is implemented since there were no human execution results for comparison...'
- 'GTPfollower1 embedded in the CB2 Platform...'

---

### Official Review · Reviewer_gzRJ · 2024-11-07

**Soundness:** 2
**Presentation:** 2
**Contribution:** 3
**Rating:** 5
**Confidence:** 4

**Summary:**

This work presents an approach for domain adaptation of language model-based agents in complex embodied tasks. To address the challenge of efficiently adapting large language models (LLMs) for domain-specific tasks while leveraging the advantages of parameter-efficient smaller models, the authors introduce the Dynamic Instruction Decomposition (DID) framework. Within this framework, an LLM decomposes high-level user instructions into immediate tasks (executable within the current context) and deferred tasks (requiring additional context). The framework progressively executes immediate tasks using external tools for low-level control, while continuously decomposing deferred tasks until completion. The authors demonstrate their approach in the CB2 collaborative environment, where they first collect expert demonstrations using an LLM integrated with the DID framework. This dataset is then augmented and filtered through a quality control process to create a high-quality training dataset. Finally, a smaller language model (sLM) is fine-tuned on this dataset and replaces the original LLM in the DID framework, enabling efficient environment interaction while maintaining the ability to handle open-ended, diverse instructions.

**Strengths:**

1. The paper addresses a promising yet under-explored direction of efficiently deploying LLMs in embodied environments.

2. The framework's approach of decomposing high-level instructions into immediate and deferred tasks provides an effective solution for handling long-horizon tasks. Moreover, delegating environmental control to external tools aligns well with current LLM research trends such as tool-use LLM and symbolic approaches.

3. The authors demonstrate clear performance improvements in a challenging collaborative environment.

**Weaknesses:**

1. The paper's presentation diminishes the generalizability of the proposed framework. While the DID framework - decomposing instructions, executing tasks, collecting datasets, and distilling knowledge - could be applicable to various embodied environments, the early introduction and focus on CB2 may lead readers to question its general applicability. A more general presentation of the framework, followed by CB2 as an implementation example, would better highlight the method's broad potential.

2. The experimental validation is limited. Testing on widely-used embodied environments such as VirtualHome[1] and Alfred[2] would better demonstrate the framework's generality.

While I believe this work holds value and has the potential to make a meaningful impact, addressing the presentation and expanding empirical validation across diverse environments would substantially strengthen the paper. I encourage the authors to consider these improvements in a revised version, as doing so could greatly enhance the clarity and general applicability of the proposed framework.

[1] VirtualHome: Simulating Household Activities via Programs, CVPR 2018

[2] ALFRED: A Benchmark for Interpreting Grounded Instructions for Everyday Tasks, CVPR 2020

**Questions:**

1. Could you clarify the baseline implementations? Specifically:
    - Does the Decision Transformer baseline use the same dataset as DID-Mistral-7b, only differing in neural architecture? If so, this comparison might be better suited for demonstrating architecture robustness rather than as a main baseline.
    - In the DID effectiveness ablation study, what are the specific implementations of GPTFollower and GeminiFollower?

2. What is the motivation behind using fine-tuning over alternative approaches? For instance, could treating the collected dataset as a knowledge base and employing RAG for dynamic prompting be effective? Please refer to [1].

3. Minor comments
    - Typos: "controllerwhich" (Line 258), "Unpon" (Line 140), "Mixtral" (Table 1).


[1] LLM-Planner: Few-Shot Grounded Planning for Embodied Agents with Large Language Models, ICCV 2023

---

### Official Review · Reviewer_yuU7 · 2024-11-09

**Soundness:** 2
**Presentation:** 2
**Contribution:** 1
**Rating:** 3
**Confidence:** 4

**Summary:**

This paper studies the problem of the instruction following, in which the agent is given an instruction to complete the task. To solve this problem, the paper proposes a method that uses a closed-source LLM to obtain the training data, which is later used to fine-tune a samller but open-sourced LLM. There are several advantages to doing this. First, the fine-fined LLM can be much more cost-effective without sacrificing the performance of tasks. Second, the fine-tuned LLM can be adapted to any downstream task. The method is called "LLMs Synergy". The method consists of two processes: it first uses a larger LLM to generate the data, and then the data is used to fine-tune LLMs. The second contribution is the method called "Dynamic Instruction Decomposition (DID)". The method divides the high-level task into smaller tasks to provide better action plans. Two types of plans (tasks) are used: immediate tasks that can be executed given the current context of the task, and deferred tasks that require more information while exploring the environment. Finally, the paper applies the method in one task, called CB2 -- a human-agent collaboration game. Extensive experiments are conducted to verify the method.

**Strengths:**

1. The core idea of the proposed method is easy to understand -- it is a distilling process to train a smaller LLM to be more cost-effective

**Weaknesses:**

1. The writing of the paper can be improved. Here are a few examples. First, in the introduction, it is very confusing at the beginning to understand the main problem that the paper wants to solve. Second, usually, the task section can be moved to the experiment section so that the reader can easily refer to the task setup. Third, section 4.1 is very wordy, it is hard to understand the core method. Fourth, the paper inverted a new term "model transition" in the introduction without actually explaining it. It takes readers to read 2 more sections to understand its definition.
2. The proposed method seems to have overfitted the CB2 task. For instance, how to apply this approach to the task in which the agent receives the visual RGB input. Second, the paper only shows the results in CB2 tasks -- it is unclear to know how general the method is.
3. There have been tons of papers proposing to use one LLM to generate data to further fine-tune the smaller LLM. For instance, see the paper Iterative Reasoning Preference Optimization, Pang et al. As a result, the method is not novel.
4. The proposed method does not achieve the best results in Table 1. In addition, it would be nice to run the original version of Gemini or GPT to understand its baseline performance.
5. For the above reasons -- lack of other tasks to verify the effectiveness of the model, the novelty of the paper, the paper does not meet the bar of ICLR

**Questions:**

See the weakness section

---

### Official Review · Reviewer_ZH1A · 2024-11-10

**Soundness:** 2
**Presentation:** 2
**Contribution:** 2
**Rating:** 3
**Confidence:** 2

**Summary:**

The authors introduce new data generation and model distillation recipes/frameworks for instruction-giving and instruction-following agents in open-ended task settings. The new contribution of these frameworks is intended to be the use of a large LLM to generate data on which smaller models can be fine-tuned for both greater task efficiency and better performance.

**Strengths:**

The paper is reasonably organized and studies an interesting task and application of LLMs.

**Weaknesses:**

Weakness A: I found it difficult to understand what the precise contributions of the paper were, and whether or not the contributions were experimentally verified to be impactful. Some key issues about this were the following:

1) What exactly is the "LLM Synergy" framework? This name was only mentioned in the first bullet of the Introduction section (L55). The authors did not follow-up to describe what this is, or how it is placed in the overall contribution.

2) In the third bullet of the introduction section, the authors claim that the DID framework "significantly improves" two things: (a) "agents' comprehension of natural language" and (b) "adaptability to dynamic environments". It was not clear to me that either of these results were verified by the experiments; the authors only benchmarked their models on the CB4 task. It would be better to see more discussion of claims (a) and (b).

3) Similarly, in the third bullet of the Introduction section, the authors write "Our experimental results demonstrate substantial improvements in...human-agent collaboration" (L67). I don't understand this -- it seems that all experiments were run under the agent-agent collaboration setting.

Weakness B: A key point that the authors return to throughout the work is how their framework increases adaptability of agents to complex tasks. However, the authors only show results on the CB4 task. To make their point better, it would be more interesting to see how the DID framework can apply to other tasks as well.

Weakness C: there are numerous typos and writing errors throughout the work.

**Questions:**

L54: It is not clear what the "LLMs Synergy" framework is, in contrast to the "Dynamic Instruction Decomposition" framework. The first paragraph of Section 4 (L136) seems to address "synergy", but all that is described are the roles that the larger and smaller LLMs play in the CB4 task. I do not understand what the authors are claiming is newly contributed.

L78: Why is the CB4 task not also "narrowly-defined"?

L118: "As a result, the descriptions within leader's instructions focus on the surroundings rather than specific details" -- aren't surroundings also specific details? I don't understand the point being made here.

L136: What is the difference between the "general LLM" and the "task-specific LLM"? It seems that the "general" LLM is the one that is eventually replaced with a fine-tuned model. What about the "task-specific" LLM? Is the same model used for each?

L140: What is the setup phase?

L383: Why is only DID-Mixtral-7b-Finetuned labelled with "(Ours)"? Isn't the DID framework a contribution of this work, and if so aren't all DID-* rows part of the contribution?



L140: "Unpon" -- typo

---

### Official Review · Reviewer_xJrL · 2024-11-10

**Soundness:** 2
**Presentation:** 2
**Contribution:** 2
**Rating:** 3
**Confidence:** 3

**Summary:**

The paper presents a novel approach for constructing efficient instruction-following agents using LLMs. The authors introduce LLMs Synergy, a method that leverages a large general-purpose LLM to establish task baselines and generate domain-specific data, which is then transferred to a smaller, domain-tuned open-source LLM. This process is facilitated by the DID framework, which aligns open-ended instructions with dynamic environmental contexts. The experimental results demonstrate significant improvements in task accuracy and human-agent collaboration, highlighting the potential of this approach for rapid domain adaptation and efficient instruction following.

**Strengths:**

Overall, the development of the DID framework for aligning natural language commands with dynamic environmental contexts is an important direction, addressing a common challenge in embodied AI. The proposed method has the potential to enhance human-agent collaboration in real-time environments, which is a critical area of research in AI.

By demonstrating how to effectively combine closed-source and open-source models, the paper opens up new possibilities for leveraging the strengths of both types of models in various domains.

**Weaknesses:**

The motivation seems strange to me: as large closed-source model can not address the problem of a specific domain well, why can we transfer its knowledge to a smaller model? On the other hand, if large models perform well, we do not need the small model any more.

Besides, I have some other issues:
1. The experimental comparisons are limited. It does not sufficiently compare the proposed method with a wide range of existing approaches. Authors can consider including more baselines, e.g., more representative LLM-Based Agents, and conducting ablation studies to isolate the impact of each component of the DID framework would strengthen the validity of the results.
2. The dataset used for training and evaluation is limited in scope. Especially, CB2 is not a representative embodied instruction following tasks. Maybe robotics control following instructions would be more suitable. Increasing the diversity and size of the dataset, and providing more details on how the data was collected and processed, would improve the reliability of the findings. Additionally, discussing potential biases in the data and how they were mitigated would be valuable.

**Questions:**

Refer to weakness section.

---

### Note · Authors · 2024-11-27

I have read and agree with the venue's withdrawal policy on behalf of myself and my co-authors.